# Evaluation of Sleep Disturbances and Depression in Children with Juvenile Idiopathic Arthritis Using the Beck Depression Inventory and Child Sleep Disorder Scale: Relationship with Leukocyte and Neutrophil Counts

**DOI:** 10.3390/children12111470

**Published:** 2025-10-31

**Authors:** Seyda Dogantan, Evin İlter Bahadur, Burcu Bozkaya Yücel, Adem Keskin, Esma Bekece

**Affiliations:** 1Department of Pediatric Rheumatology, Basaksehir Cam and Sakura City Hospital, Istanbul 34480, Turkey; 2Institute of Graduate Studies in Health Sciences, Istanbul University, Istanbul 34134, Turkey; 3Department of Developmental Pediatrics, Mersin City Hospital, Mersin 33240, Turkey; evinbahadur@gmail.com; 4Department of Pediatric Rheumatology, Samsun Research and Education Hospital, Samsun 55090, Turkey; bozkayaburcu@hotmail.com; 5Department of Medical Biochemistry, Faculty of Medicine, Aydin Adnan Menderes University, Aydin 09100, Turkey; adem.keskin@adu.edu.tr; 6Department of Pediatrics, Basaksehir Cam and Sakura City Hospital, Istanbul 34480, Turkey; ebekece@gmail.com

**Keywords:** juvenile idiopathic arthritis, beck depression inventory, sleep disturbance scale for children, leukocytosis, erythrocyte sedimentation rate, leukocyte, red cell distribution width, neutrophil, vitamin D, C-reactive protein

## Abstract

**Highlights:**

**What are the main findings?**
•Children with JIA had higher BDI and SDSC scores and SDSC subscales (except SHY) compared to healthy children.•BDI and SDSC scores in children with JIA showed a positive relation with each other and a negative relation with leukocyte and neutrophil counts.

**What are the implications of the main findings?**
•The risks of depression and sleep disorders should be considered in patients with JIA.•Sleep disorders and depression may trigger each other in patients with JIA.

**Abstract:**

**Background/Objectives**: The pathophysiology of juvenile idiopathic arthritis (JIA), the most widespread rheumatologically illness in juvenile period, is shaped by complex interactions between leukocytes and the cytokines they secrete. The aim of this research was to evaluate the severity of sleep disturbances and depression, which are closely associated with many diseases and can negatively impact the course of the illness, in patients with JIA using Beck Depression Inventory (BDI) and Sleep Disturbance Scale for Children (SDSC) scores and to investigate the relationship between these scores and laboratory findings in patients with JIA. **Methods**: The research involved 58 children with JIA and 71 healthy children as controls. BDI and SDSC scores of these groups were compared with laboratory findings and correlation analysis were performed. **Results**: In the JIA group, BDI and SDSC scores, C-reactive protein (CRP), red blood cell distribution width (RDW), erythrocyte sedimentation rate, neutrophil, and leukocyte counts, were higher than in the control group, while vitamin D values were lower. A positive relation was determined between BDI and SDSC scores in the JIA group, but no correlation was found in the control group. In the JIA group, both BDI and SDSC scores were found to be negatively related with leukocyte and neutrophil counts. In the control group, the BDI score was determined to be negatively correlated with CRP, vitamin D and RDW levels. **Conclusions**: Depression and sleep disorders may interact in patients with JIA, and their causal relationship with leukocyte and neutrophil levels should be investigated.

## 1. Introduction

Juvenile idiopathic arthritis (JIA), the most widespread rheumatologic illness in childhood, is a chronic rheumatic illness described by progressive joint destruction and severe systemic manifestations. The pathophysiological process is mediated by complex interactions between immune cells such as lymphocytes, monocytes, macrophages, and neutrophils [1]. While the precise etiology of JIA is not yet understood, inflammation in the synovium and destruction of joint tissues play a key role, particularly in the early onset of the oligoarticular subtype. Clinically, the disease typically presents with joint pain and swelling, and the lack of specific laboratory tests to confirm the diagnosis often makes diagnosis challenging [2]. Given the heterogeneous nature of JIA, which constitutes a group of chronic inflammatory arthropathies, a multidisciplinary and comprehensive approach is required for its treatment. In recent years, there has been a clear trend toward targeted treatment approaches in the management of JIA [3]. Therefore, there is a need to elucidate the complex pathophysiological mechanisms of JIA in more detail in order to facilitate early diagnosis of the illness and direct the therapy process.

Depression is a common illness that negatively impacts life quality and poses an important economic burden on public. There is growing evidence that depression is closely linked to other illnesses and may negatively impact their outcomes [4,5,6,7,8,9]. Furthermore, the prevalence of depression is declared to be higher in rheumatology cases compared to the frequency population at large. Socioeconomic factors do not adequately explain mood disorders in these cases. The clinical features declared by cases with chronic inflammatory illnesses resemble changes described as “sickness behavior” [10]. On the other hand, in the assessment of depression, the Beck Depression Inventory (BDI) stands out as a reliable and valid measurement tool widely used in determining the presence and severity of depression both in clinical practice and in research [11].

Sleep is a vital biological process necessary for maintaining both healthy neurological functions and overall health. However, a significant portion of the population experiences various sleep disorders that often go unnoticed or underrecognized. This reduces individuals’ quality of life and can have long-term negative effects on physical and mental health [12]. Sleep disorders, which sometimes arise due to lifestyle or work conditions, or sometimes due to illness, not only reduce an individual’s quality of life but are also closely linked to illnesses such as depression and can negatively impact the course of these illnesses [12,13,14,15,16,17]. However, the Sleep Disturbance Scale for Children (SDSC) is a multidimensional and reputable measure developed to assess sleep disorders in children, completed by caregivers; it covers areas such as difficulty falling asleep, breathing-related problems, and waking disorders [18].

JIA, whose pathophysiological process progresses through complex interactions between leukocytes and the macrophages they differentiate into, is a heterogeneous disease requiring a multidisciplinary and comprehensive treatment approach. Furthermore, depression and sleep disorders, which are closely associated with many diseases and can negatively impact the course of the disease, are reliably assessed using the BDI and SDSC scores. The aim of this research was to evaluate the severity of sleep disorders and depression in children with JIA using the BDI and SDSC scores and to examine the potential impact of these disorders on the course of the disease. It also aimed to examine the relation between these scores and laboratory findings in children with JIA.

## 2. Materials and Methods

### 2.1. Research Population

The number of children involved in the research was determined using the G*Power 3.1.9.7 power analysis program (Test family: *t* tests, Statistical test: Means: Difference between two independent means (two groups), Type of power analysis: A priori: Compute required sample size—given α, power, and effect size, (Effect size d = 0.6, α = 0.05, power (1 − β) = 0.95, Allocation ratio N2/N1: 0.817 (58/71)). As a result of the power analysis, the minimum number of participants required for the JIA group was determined as 56 and for the control group as 68. Therefore, fifty-eight children diagnosed with JIA who were followed in the Pediatric Rheumatology Clinic of Başakşehir Çam and Sakura Hospital were involved in the research as the JIA group. All individuals included in this group were in remission. JIA diagnoses were established according to the ILAR (International League Against Rheumatism) classification criteria [19,20]. In addition, the diagnosis was confirmed using the recently proposed Pediatric Rheumatology International Trials Organization (PRINTO)/EULAR (European Alliance of Associations for Rheumatology)/ACR (American College of Rheumatology) classification criteria [21,22]. Patient exclusion criteria included concurrent chronic diseases, including other autoimmune disorders, and previous use of glucocorticoids or other immunosuppressive agents. In addition, seventy-one healthy children without malignancy, chronic illness, inflammatory or hematological disorders, or medication use were involved in the research as the control group. In this research, individuals under the age of 13 administered depression and sleep disturbance scores with parent/guardian support to increase the clarity and reliability of their responses. For children in this age group, the questionnaires were completed by their parents or guardians. During administration, the researchers guided parents to respond to each item based on their child’s recent emotional state and behavior. This approach ensured the reliability and consistency of the data obtained from young children.

### 2.2. Beck Depression Inventory (BDI) Score Assessment

BDI is a commonly used self-report measure developed to evaluate the severity and presence of depression-related clinical features. It consists of 21 items, each with four options scored from 0 to 3. The total score, obtained by summing the scores of all items, ranges from 0 to 63, with higher scores indicating more severe depressive symptoms. Commonly used classification systems include scores of 0–13 indicating minimal or no depression, 14–19 indicating mild depression, 20–28 indicating moderate depression, and 29–63 indicating severe depression. BDI is frequently used in clinical practice to assess depression, monitor treatment, and compare patient groups to healthy controls in research [23].

### 2.3. Sleep Disturbance Scale for Children (SDSC) Score Assessment

SDSC is a validated parent-reported questionnaire developed to assess sleep problems in kids and adolescents aged 6–15 years. It consists of 26 items, each rated on a 5-point Likert scale (ranging from “always” to “never”), that evaluate the frequency and intensity of sleep-related difficulties during the previous 6 months. The SDSC provides a total score as well as scores for 6 subscales: (1) DIMS (disorders of initiating and maintaining sleep), (2) SBD (sleep breathing disorders), (3) DA (disorders of arousal), (4) SWTD (sleep–wake transition disorders), (5) DOES (disorders of excessive somnolence), and (6) SHY (sleep hyperhidrosis). Higher scores indicate more severe sleep disturbances. The SDSC has been widely used in both research and clinical settings as a reliable tool to identify and monitor sleep disorders in the child population [24]. In addition, in the research by Bruni et al., individuals with a total SDSC score ≥ 39 were in the upper quartile, and this value is recommended as an indicator of sleep disturbance, with a specificity of 0.74 and a sensitivity of 0.89 [25]. In the literature, an SDSC score ≥ 39 is interpreted as an indicator of poor sleep quality [26,27,28].

### 2.4. Statistical Analysis

SPSS 22 for Windows (IBM, New York, NY, USA) was used to analyze statistical data. The conformity of continuous variables to normal distribution was assessed using skewness and kurtosis values and the Shapiro–Wilk test. Continuous data with normal distribution are shown as mean ± standard deviation (X ± SD), and continuous data without normal distribution are shown as median (25th and 75th percentile ranks). Categorical data are shown as n (percentage frequency). One-way Anova test and Independent Samples *t*-test were used among parametric tests to compare continuous data with normal distribution. Mann–Whitney U test and Kruskal–Wallis test was used among nonparametric tests to compare continuous variables without normal distribution. Pearson correlation test was used in correlation analysis. The chi-square test was used to compare categorical data. *p* < 0.05 was accepted as the limit of statistical significance. To control for false positive results that may arise from multiple comparisons, Holm–Bonferroni correction was applied to the intergroup comparisons of biomarkers and SDSC subscales.

## 3. Results

Fifty-eight children diagnosed with JIA between the ages of 4 and 18 were involved in the JIA group of the study. The control group encompassed of 71 healthy child aged 6–16 years. The mean age of the JIA group was 12.36 ± 4.27, while the mean age of the control group was 9.18 ± 2.50. The mean age of the JIA group was higher than the mean age of the control group (*p* < 0.001, Independent samples *t*-test). In the JIA group, 55.17% (n = 32) were female and 44.83% (n = 26) were male. In the control group, these rates were 56.34% (n = 40) and 43.66% (n = 31), respectively. No important difference was determined between the two groups in terms of gender ratios (*p* = 0.894, Chi-square test). The average depression and sleep disturbance scores of these two groups are presented in Table 1.

The BDI, total SDSC and SDSC subscale data of the groups were compared using an independent samples *t*-test (with Holm-Bonferroni correction). Additionally, the categorical distribution of depression severity and poor sleep quality between groups was compared using a chi-square test. Mean BDI, total SDSC and SDSC subscales (except SHY) were higher in the JIA group than in the control group (Table 1). Furthermore, an important difference was found between the distributions of depression severity categories and poor sleep quality in the two groups (Table 1). Interestingly, the SHY subscale mean was lower in the JIA group than in the control group (Table 1).

Correlation analysis was tested to examine the relationship between the BDI score, SDSC score, and SDSC subscales in both groups (Table 2).

While a positive relation was found between the BDI and the SDSC scores and SDSC subscales (except SHY) in the JIA group, no correlation was determined in the control group (Table 2). One the other hand, a negative relation was determined between the BDI and SHY subscales in the JIA group, no correlation was found in the control group (Table 2).

Laboratory results of the JIA and control groups are presented in Table 3.

The erythrocyte sedimentation rate, serum amyloid A, mean platelet volume, neutrophil/lymphocyte ratio, folate, ferritin, leucocyte count, C-reactive protein, red blood cell distribution width, lymphocyte, and neutrophil values of the groups were compared using a Mann–Whitney U test (with Holm–Bonferroni correction). Additionally, the hemoglobin, platelet, vitamin D, and vitamin B12 values of the groups were compared using an independent samples *t*-test (with Holm–Bonferroni correction). While erythrocyte sedimentation rate, leukocyte count, C-reactive protein, red cell distribution width, and neutrophil values were determined to be higher in the JIA group than in the control group, vitamin D values were determined to be lower in the JIA group than in the control group (Table 3). Correlation analysis was tested to see the relationship between these significant laboratory findings and BDI and SDSC scores in both groups (Table 4).

In the JIA group, both BDI and SDSC scores were found to be negatively correlated with leukocyte and neutrophil levels, but not with other laboratory findings (Table 4). In the control group, the BDI score was determined to be negatively related with CRP, vitamin D and RDW values, and not significantly correlated with other laboratory findings (Table 4).

In the JIA group, 18.97% (n = 11) of patients had enthesitis-related arthritis, 41.38% (n = 24) had oligoarticular JIA, 24.14% (n = 14) had polyarticular JIA, 6.90% (n = 4) had psoriatic arthritis, and 8.62% (n = 5) had systemic JIA. When the JIA group was classified according to disease subtypes and mean BDI and SDSC scores were compared, no important differences were determined between the subgroups (*p* = 0.464, *p* = 0.634, respectively, One-Way Anova). Similarly, no important differences were determined in laboratory findings. The erythrocyte sedimentation rate, serum amyloid A, mean platelet volume, neutrophil/lymphocyte ratio, folate, ferritin, leucocyte count, red blood cell distribution width, C-reactive protein, lymphocyte, and neutrophil values of the groups were compared using a Kruskal–Wallis test. Additionally, the hemoglobin, platelet, vitamin D, and vitamin B12 values of the groups were compared using a One-Way Anova test.

## 4. Discussion

Depression, which disproportionately affects individuals with chronic illness, is a serious health problem that is often overlooked and underrecognized in routine care in child rheumatology clinics. A recent research reported that depressive symptoms in patients with JIA were associated with rheumatoid factor-negative polyarthritis, increased pain scores, functional limitations, high illness activity, poor public health, increased medication use, and failure to achieve remission. Furthermore, persistent pain despite treatment and minimal disease activity or failure to achieve remission despite biologic therapies were also strongly related with depressive symptoms [29]. A research of 145 children with JIA assessed depression using the Hamilton Scale and found that one-third of children with JIA reported depression. Furthermore, symptoms of depression were found to be related with illness activity [30]. A recent study of 1150 adolescents with JIA reported a high frequency of depression and anxiety symptoms in JIA, highlighting the need for routine monitoring for early detection of mental health issues [31]. A recent study of patients with psoriatic arthritis reported that delays of more than six months between symptom onset and diagnosis increased the likelihood of depression, with longer delays leading to progressively worse outcomes [32].

Most patients with JIA involved in this research had mild, moderate, or severe depression as assessed by the BDI. Furthermore, the mean BDI scores of participants with JIA were higher than those of healthy kids.

Sleep disturbances in adolescents and kids with JIA arise from a variety of clinical features and mechanisms, with subclinical inflammation exacerbating clinical symptoms and disease progression, leading to a vicious cycle. Possible contributing factors to sleep disturbances include chronic inflammation, disease symptoms, psychiatric comorbidities, and circadian rhythm disturbances that may emerge during adolescence [33]. Furthermore, Tarakçı et al. reported that sleep challenges are widespread in JIA and are related with disease activity, pain, functional impairment, and fatigue. They also reported poor sleep quality in 40% of the study population [34]. Additionally, sleep disorders are one of the key characteristics of psychiatric disorders, including depression. The relationship between sleep and depression is complex, bidirectional, and difficult to understand. Indeed, it has been repeatedly demonstrated that both objective and subjective measures of sleep disturbance increase in depression. The co-occurrence of sleep disorders and depressive mood causes a significant deterioration in the life quality of affected individuals [35].

Two-thirds of the patients with JIA included in this research had poor sleep quality. Furthermore, the SDSC score and SDSC subscale (except SHY) scores were higher in the children with JIA than in healthy children. In addition, both the SDSC score and SDSC subscale (except SHY) scores were positively related with the BDI score. One the other hand, SDSC SHY subscale score were lower in the participants with JIA than in healthy children. Furthermore, SDSC SHY subscale scores were negatively correlated with the BDI score. However, both sleep hyperhidrosis and depressive symptoms are related with overactivation of the sympathetic nervous system, suggesting that depressive symptoms play a more dominant role in sympathetic nervous system activity by suppressing sleep hyperhidrosis [36,37]. Additionally, children with JIA included in this study were under follow-up and in remission. The negative correlation between sleep hyperhidrosis and BDI scores is thought to be due to the anti-inflammatory impacts of the medications used. The treatment’s ability to control inflammation provides a plausible basis for the indirect reduction in symptoms such as sleepiness and sweating [38]. Indeed, a study by Khubchandani and colleagues reported improvement in sleep hyperhidrosis after treatment in children with active disease [39].

JIA is one of the leading rheumatic illnesses in kids, mediated by both the adaptive and innate immune systems. Vitamin D can enhance the endocrine activity and immunomodulatory activity of monocytes and macrophages. Given the frequency of vitamin D deficiency in participants with JIA and the potential for vitamin D to influence bone mineralization and immune modulation, the clinical importance of vitamin D in these individuals is clearly established [40]. Moreover, a recent research reported that a high degree of disease activity in JIA was related with low vitamin D levels [41]. Additionally, in a study evaluating 62,835 vitamin D results indiscriminately in children and adults, values below 20 ng/mL were determined as vitamin D deficiency, and values between 20 and 30 ng/mL were determined as suboptimal vitamin D values (vitamin D insufficiency) [42]. A recent update study on vitamin D deficiency supports these definitions [43].

Vitamin D insufficiency and deficiency have been detected in children participating in this study, as reported on a global scale. This study found that the mean vitamin D values in participants with JIA were below the vitamin D deficiency threshold, while vitamin D values in healthy participants were below optimal levels. Furthermore, mean vitamin D values in participants with JIA were found to be lower than in healthy children. On the other hand, while there was a negative relation between vitamin D values and BDI scores in healthy children, no important correlation was determined between vitamin D values and BDI or SDSC scores in children with JIA.

CRP, ESR, and white blood cell counts are parameters traditionally used for both patient monitoring and assessing disease activity in JIA, and they are also used as a reference for comparison in the investigation of the value of new biomarkers for JIA [44,45,46,47].

In this study, CRP, leukocyte, ESR, and neutrophil count were found to be higher in participants with JIA than in healthy children. Although these values are statistically higher in participants with JIA compared to healthy participants, the magnitude of the difference between the two-group averages can be considered clinically small. This may be due to the fact that the patients with JIA included in the research were in remission and under regular follow-up. In addition, both BDI and SDSC scores were found to be highly negatively correlated with neutrophil and leukocyte counts in children with JIA. However, no significant relation was determined between these scores and ESR and CRP levels in children with JIA. This suggests that this may be due to the treatment administered to these children during remission. Because CRP and ESR levels are expected to remain within normal limits during this period, the treatment administered may have obscured a possible correlation. Furthermore, because immunomodulatory or immunosuppressive treatments (e.g., methotrexate, corticosteroids, or biologic agents) suppress the immune system in children with JIA during remission, it is possible that white blood cell counts were in the lower limit of normal or slightly decreased during this period. This may partially explain the negative correlation observed between leukocyte and neutrophil counts and these scores.

A recent meta-analysis reported a consistently positive association between mental health and sleep in both adolescents and children. It also suggested that increased sleep may have a protective effect on mental health [48]. Conversely, insufficient sleep in children have been reported to lead to a variety of neurological, neurocognitive, emotional, and mood-related problems. However, sleep duration is an age-dependent variable; children of different ages require different amounts of sleep. Adequate sleep quality can be defined as an individual’s ability to maintain a normal sleep pattern appropriate for their age. Therefore, the patient’s age and associated etiological factors should be taken into account when planning treatment for sleep insufficiency [49]. The healthy children involved in this research were younger than the children with JIA.

When children with JIA were divided into subgroups according to disease subtypes and compared in terms of BDI and SDSC scores as well as laboratory findings, no significant differences were observed among the subgroups. However, considering the heterogeneous nature of JIA, the sample size can be regarded as a limitation of the study. This sample size limitation restricts the analysis of gender-specific differences in the relationships between depressive and sleep disorder symptoms and JIA, despite the absence of an important difference in the observed gender ratios between participants with JIA and healthy participants. Future research with larger sample sizes could address this gap by analyzing both potential sex-specific differences and potential differences between JIA subtypes. Additionally, lack of data on medication use and disease duration, age differences between groups, absence of patients in the active phase, potential self-report bias, cross-sectional study design preventing causal inference, and lack of psychological support information for participants are among the important limitations of the study. On the other hand, according to the literature review, this research represents the first to simultaneously assess depression and sleep disturbances in children with JIA and compare them with laboratory findings, which constitutes a major strength. Moreover, the results provide new perspectives on the complex pathophysiological mechanisms of JIA, patient follow-up, prognosis assessment, and therapeutic approaches.

## 5. Conclusions

In conclusion, children with JIA exhibited higher scores for depression and sleep disorders compared to healthy controls, and these two health problems were positively correlated with each other and negatively correlated with leukocyte and neutrophil levels. These findings highlight the importance of routine screening and early management of depression and sleep disorders in JIA to improve both mental health and disease outcomes.

## Figures and Tables

**Table 1 children-12-01470-t001:** Mean depression and sleep disturbance scores of the groups.

Parameters		JIA (n = 58)	Control (n = 71)	*p* *
Beck Depression Inventory X ± SD		11.48 ± 6.11	7.51 ± 4.67	<0.001
Depression severity categories n (%)	None/minimal	14 (24.14)	38 (53.52)	<0.001
	Mild	17 (29.31)	32 (45.07)	
	Moderate	16 (27.59)	1 (01.41)	
	Severity	11 (18.97)	0	
Total SDSC Score X ± SD		52.93 ± 17.34	37.61 ± 4.77	<0.001
Poor sleep quality (SDSC score ≥ 39) n (%)		40 (68.97)	18 (25.35)	<0.001
Subscales X ± SD	1-DIMS score	13.83 ± 5.09	9.93 ± 2.25	<0.001
	2-SBD score	6.59 ± 2.51	4.97 ± 1.06	<0.001
	3-DA score	6.68 ± 2.28	4.96 ± 0.87	<0.001
	4-SWTD score	11.64 ± 5.43	7.41 ± 0.82	<0.001
	5-DOES score	10.21 ± 4.33	6.79 ± 1.31	<0.001
	6-SHY score	1.95 ± 0.78	3.55 ± 0.73	<0.001

* For normally distributed continuous data presented with X ± SD: Independent Samples *t*-Test (with Holm-Bonferroni correction), for categorical variables presented with n (%): Chi-square test. JIA: Juvenile Idiopathic Arthritis, X ± SD: Mean ± Standard Deviation, SHY: Sleep hyperhidrosis, DOES: Disorders of excessive somnolence, DA: Disorders of arousal, SWTD: Sleep–wake transition disorders, SBD: Sleep breathing disorders, DIMS: Disorders of initiating and maintaining sleep, SDSC: Sleep Disturbance Scale for Children.

**Table 2 children-12-01470-t002:** Correlation analysis of BDI scores, SDSC scores, and SDSC subscales in both groups.

Parameters		JIA (n = 58)		Control (n = 71)	
		Beck Depression Inventory			
		r	*p* *	r	*p* *
Total SDSC Score		0.673	<0.001	−0.054	0.657
Subscales	1-DIMS score	0.537	<0.001	−0.204	0.089
	2-SBD score	0.575	<0.001	−0.141	0.242
	3-DA score	0.617	<0.001	0.090	0.457
	4-SWTD score	0.625	<0.001	0.014	0.906
	5-DOES score	0.517	<0.001	0.079	0.515
	6-SHY score	−0.296	0.024	0.085	0.482

JIA: Juvenile Idiopathic Arthritis, r: Correlation coefficient, *: Pearson correlation test, SHY: Sleep hyperhidrosis, SWTD: Sleep–wake transition disorders, DOES: Disorders of excessive somnolence, SBD: Sleep breathing disorders, DA: Disorders of arousal, DIMS: Disorders of initiating and maintaining sleep, SDSC: Sleep Disturbance Scale for Children.

**Table 3 children-12-01470-t003:** Laboratory results of the groups.

Parameters *	JIA (n = 58)	Control (n = 71)	*p* *
CRP (mg/dL)	5 (1–29)	2.9 (0.7–6.5)	0.007
ESR (mm/hour)	12 (4–25)	5 (3–9)	0.007
Serum Amyloid A	1.75 (0.3–7)	1 (1–3)	0.652
Leucocyte (10^3^/µL)	8.83 (7.09–11.68)	7.2 (6.1–8.74)	0.001
Hemoglobin (g/dL)	12.29 ± 1.31	12.69 ± 1.07	0.059
RDW (%)	14 (13–16)	12.4 (11.8–13.2)	<0.001
Platelet (10^3^/µL)	362,775 ± 97,277	352,845 ± 89,147	0.547
Mean platelet volume (fL)	10.5 (9.9–11.1)	10.6 (9.7–11.8)	0.632
Neutrophil (10^3^/µL)	4.75 (3.24–6.71)	3.6 (3.13–4.28)	0.012
Lymphocyte (10^3^/µL)	3.1 (2.38–4)	2.59 (2.16–3.41)	0.062
Neutrophil/Lymphocyte ratio	1.53 (0.96–2.23)	1.35 (0.99–1.97)	0.377
Vitamin D (ng/mL)	16.45 ± 7.53	20.01 ± 7.29	0.007
Vitamin B12 (pg/mL)	302.45 ± 75.61	282.01 ± 69.15	0.112
Folate (ng/mL)	7.35 (5.37–9.15)	7.26 (5.23–8.58)	0.473
Ferritin (ng/mL)	27.5 (17–45)	34 (23.5–45)	0.238

* For normally distributed continuous data presented with X ± SD: Independent Samples *t*-Test (with Holm-Bonferroni correction), for continuous data that are not normally distributed and are presented with medians (25th and 75th percentile ranks): Mann–Whitney U Test (with Holm–Bonferroni correction). JIA: Juvenile Idiopathic Arthritis, CRP: C-reactive protein, RDW: Red Cell Distribution Width, ESR: Erythrocyte Sedimentation Rate.

**Table 4 children-12-01470-t004:** Correlation analysis of BDI scores, SDSC scores, and laboratory findings in both groups.

Parameters	JIA Group (n = 58)				Control Group (n = 71)			
	BDI Score		SDSC Score		BDI Score		SDSC Score	
	r	*p* *	r	*p* *	r	*p* *	r	*p* *
ESR	0.023	0.864	−0.021	0.876	−0.210	0.079	0.121	0.313
CRP	−0.137	0.305	−0.124	0.353	0.545	<0.001	0.197	0.100
Leukocyte	−0.526	<0.001	−0.815	<0.001	−0.213	0.075	−0.062	0.606
RDW	−0.067	0.617	−0.024	0.858	−0.370	0.001	0.120	0.319
Neutrophil	−0.401	<0.001	−0.686	<0.001	−0.136	0.259	0.055	0.649
Vitamin D	0.015	0.911	−0.169	0.206	−0.327	0.005	0.014	0.907

JIA: Juvenile Idiopathic Arthritis, r: Correlation coefficient, *: Pearson correlation test, SDSC: Sleep Disturbance Scale for Children, BDI: Beck Depression Inventory, CRP: C-reactive protein, RDW: Red Cell Distribution Width ESR: Erythrocyte Sedimentation Rate.

## Data Availability

The data are not publicly available due to confidentiality or ethical restrictions. Additionally, The datasets used and/or analyzed during the current study are available from the corresponding author on reasonable request.

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
