# Peer review of "Evaluation of Sleep Disturbances and Depression in Children with Juvenile Idiopathic Arthritis Using the Beck Depression Inventory and Child Sleep Disorder Scale: Relationship with Leukocyte and Neutrophil Counts"

_children, 2025, doi:10.3390/children12111470_

Round 1
Reviewer 1 Report
Comments and Suggestions for Authors
Thank you for the opportunity to review the manuscript, Evaluation of depression and sleep disturbances in children with Juvenile Idiopahtic Arthritis using the Beck Depression Inventory and Child Slee Disorder Scale: Relationship with Laboratory Findings. The paper is well-written and its focus, the association of depression and sleep disorders with children with JIA, will be of great interest to readers of the Children —and timely, given the attention that depression and sleep dysfunction has recently received in relation to health outcomes of other disorders. Most importantly, the paper describes a well-supported experimental design comparing children diagnosed with JIA to a control group of healthy children by utilizing diagnostic surveys for depression and sleep disturbances and measuring clinical factors, including CRP, RDW, erythrocyte, leucocyte, neutrophil and Vitamin D levels. The authors find that the JIA group has a higher incidence of depression and sleep disturbance, that depression and sleep disturbance in this group are correlated and that the JIA group had increased CRP, RDW, erythrocyte, leucocyte and neutrophil levels and decreased levels of Vitamin D. Below, I outline a few comments that I hope will help strengthen the impact of the manuscript.
- A major strength of the manuscript is the straightforward design and use of well-supported/recognized diagnostic tools for measuring depressive and sleep disturbance symptoms (the BDI and SDSC). The comparison of self-reported mental health measures and the measurement of clinical factors (CRP, RDW, erythrocyte, leucocyte, neutrophil and Vitamin D levels) in JIA and control children allows for a novel assessment of how these clinical factors associate with depressive and sleep disorder symptoms in children with JIA.
- Less clear is the directionality of the association of depression and sleep disorders with JIA. The authors imply that having depression or dysfunctional sleep worsens health outcomes of children with JIA, but this is not clear from a simple association study. For example, one of the potential molecular mechanisms underlying depression involves inflammation pathways, particularly in females. So, it is not clear whether the immune system dysfunction is driving depression or whether depressive symptoms worsen immune system function. The authors explanation of the positive and negative correlations of laboratory measures with BDI and SDSC is not clear…please clarify this discussion and its relevance to the etiology of JIA.
- Recently, significant sex-specific differences have been reported for mechanisms underlying mental health, including depression, anxiety, addiction and dementia. So, the fact that this study does not investigate differences between genders is a limitation. The limited sample size is addressed as a shortcoming for the comparisons of sub-types of JIA, but it is also a limitation for the analysis of sex-specific differences in associations between depressive and sleep disorder symptoms and JIA. Future studies should address this gap.
- The statistical analysis is sound, but the authors should adjust their p-values for the multiple comparisons that they test between the JIA and control groups.
- Some of the differences between the JIA and control groups are significant but quite small (e.g. leucocyte levels), including the Vitamin D levels that are discussed at length. Can the authors provide some context on biologically relevant differences? Particularly for Vitamin D levels and the statement that they are below the healthy threshold of 20 (is this from a study of children or adults?)…Also, the healthy cohort also has a level of 20…what does this signify?
Minor comments:
- lines 70-71, 83-86 and 92-94 are repetitive.
Author Response
Reviewer 1
Thank you for the opportunity to review the manuscript, Evaluation of depression and sleep disturbances in children with Juvenile Idiopahtic Arthritis using the Beck Depression Inventory and Child Slee Disorder Scale: Relationship with Laboratory Findings. The paper is well-written and its focus, the association of depression and sleep disorders with children with JIA, will be of great interest to readers of the Children —and timely, given the attention that depression and sleep dysfunction has recently received in relation to health outcomes of other disorders. Most importantly, the paper describes a well-supported experimental design comparing children diagnosed with JIA to a control group of healthy children by utilizing diagnostic surveys for depression and sleep disturbances and measuring clinical factors, including CRP, RDW, erythrocyte, leucocyte, neutrophil and Vitamin D levels. The authors find that the JIA group has a higher incidence of depression and sleep disturbance, that depression and sleep disturbance in this group are correlated and that the JIA group had increased CRP, RDW, erythrocyte, leucocyte and neutrophil levels and decreased levels of Vitamin D. Below, I outline a few comments that I hope will help strengthen the impact of the manuscript.
Comments 1: A major strength of the manuscript is the straightforward design and use of well-supported/recognized diagnostic tools for measuring depressive and sleep disturbance symptoms (the BDI and SDSC). The comparison of self-reported mental health measures and the measurement of clinical factors (CRP, RDW, erythrocyte, leucocyte, neutrophil and Vitamin D levels) in JIA and control children allows for a novel assessment of how these clinical factors associate with depressive and sleep disorder symptoms in children with JIA.
Less clear is the directionality of the association of depression and sleep disorders with JIA. The authors imply that having depression or dysfunctional sleep worsens health outcomes of children with JIA, but this is not clear from a simple association study. For example, one of the potential molecular mechanisms underlying depression involves inflammation pathways, particularly in females. So, it is not clear whether the immune system dysfunction is driving depression or whether depressive symptoms worsen immune system function. The authors explanation of the positive and negative correlations of laboratory measures with BDI and SDSC is not clear…please clarify this discussion and its relevance to the etiology of JIA.
Response 1: Dear Reviewer,
First of all, thank you very much for taking the time to evaluate our manuscript and for your constructive feedback.
Taking your criticisms into account, the explanations regarding the positive and negative correlations of laboratory measurements with BDI and SDSC were revised more clearly, and the relationship with JIA etiology was clarified.
Comments 2: Recently, significant sex-specific differences have been reported for mechanisms underlying mental health, including depression, anxiety, addiction and dementia. So, the fact that this study does not investigate differences between genders is a limitation. The limited sample size is addressed as a shortcoming for the comparisons of sub-types of JIA, but it is also a limitation for the analysis of sex-specific differences in associations between depressive and sleep disorder symptoms and JIA. Future studies should address this gap.
Response 2:
Taking your suggestion into consideration, the following sentence has been added to the article.
“This sample size limitation restricts the analysis of gender-specific differences in the relationships between depressive and sleep disorder symptoms and JIA, despite the absence of a significant difference in the observed gender ratios between participants with JIA and healthy participants. Future research with larger sample sizes could address this gap by analyzing both potential sex-specific differences and potential differences between JIA subtypes.”
Comments 3:
The statistical analysis is sound, but the authors should adjust their p-values for the multiple comparisons that they test between the JIA and control groups.
Response 3:
To control for false positive results that may arise from multiple comparisons, Holm-Bonferroni correction was applied to the intergroup comparisons of biomarkers and SDSC subscales; however, a statement regarding this was not included in the statistical analysis section. In line with your suggestion, the following sentence regarding this issue has been added to the statistical analysis section.
“To control for false positive results that may arise from multiple comparisons, Holm-Bonferroni correction was applied to the intergroup comparisons of biomarkers and SDSC subscales.”
Comments 4:
Some of the differences between the JIA and control groups are significant but quite small (e.g. leucocyte levels), including the Vitamin D levels that are discussed at length. Can the authors provide some context on biologically relevant differences? Particularly for Vitamin D levels and the statement that they are below the healthy threshold of 20 (is this from a study of children or adults?)…Also, the healthy cohort also has a level of 20…what does this signify?
Response 4:
Based on your recommendations, biologically meaningful differences between the JIA and control groups were assessed. The relevant statement was explained in detail and supported by the literature. The significance of the vitamin D levels observed in the healthy cohort was also noted.
Comments 5:
lines 70-71, 83-86 and 92-94 are repetitive.
Response 5:
Sentences 70-71 have been edited to clarify their different meanings. In this sentence, the subject is depression.
In sentences 83-86, the subject is sleep disorder.
Sentences 92-94 summarize the introductory information regarding the purpose and essence of the study in the purpose paragraph.
Sincerely

Reviewer 2 Report
Comments and Suggestions for Authors
1. General Assessment
The article is well written, logically consistent, and aligns with the scope of the journal Children. The authors address a current and interdisciplinary topic - the relationship between depressive symptoms, sleep disturbances, and immunological markers in juvenile idiopathic arthritis (JIA). The study combines clinical, psychological, and biochemical aspects, which is undoubtedly its strength.
The research is carefully designed (power calculated in G*Power, appropriate control groups, bioethics committee approval). The conclusions are consistent with the data and literature.
However, the paper has several important limitations and areas requiring clarification, particularly in the statistical methods, interpretation of results, and linguistic accuracy.
2. Strengths
- A novel and clinically important topic - few studies simultaneously assess depression, sleep, and inflammatory parameters in children with JIA.
- Appropriate choice of diagnostic tools - BDI and SDSC are valid and well-validated instruments.
- Well-developed Materials and Methods section, especially Section 2.1, which includes the power calculation.
- Results are consistent with international literature and include numerous up-to-date references (2023–2025).
- The discussion section is extensive and effectively contextualizes the findings in relation to previous studies.
3. Substantive Comments
Methodological Design
• There is no information about the remission or activity status of JIA at the time of the study - this is crucial, as CRP, ESR, and depression severity may depend on inflammatory activity.
• The BDI scale is not suitable for younger children (under 13 years old). Participants were aged 4–18, so the tool is inappropriate for part of the sample.
It should be indicated whether parents/guardians completed the questionnaire for younger children and how response reliability was ensured.
• The control group is somewhat younger (mean age 9.18 vs. 12.36 years), which may influence sleep and mood.
This difference should be noted in the Limitations section as a potential confounding factor.
Statistical Analysis
• The authors mention using parametric and nonparametric tests but do not specify which variables were analyzed with which test (often only “p < 0.05” is provided).
Please specify for key analyses which tests were parametric and which were nonparametric.
• No correction for multiple comparisons was applied, despite analyzing several SDSC subscales and biomarkers.
Please apply Bonferroni or Holm–Bonferroni correction and update the p-values accordingly.
• The correlation strength in Table 4 appears suspiciously high and likely results from a typographical error or data duplication (r = –0.998; 0.911; 0.907; 0.876, etc.).
Interpretation of Results
• The authors interpret the negative correlation between leukocytes/neutrophils and depression as “depression reduces immunological parameters.” In a cross-sectional study, the direction of the relationship cannot be established.
Please revise such formulations.
• In the discussion, it would be valuable to expand on why SHY (sleep hyperhidrosis) was lower in the JIA group, rather than merely speculating about sympathetic activity. This may result from medication use (e.g., NSAIDs or biologics).
• The title suggests that “laboratory findings” were analyzed - it should be emphasized in the conclusion that the strongest correlations involved leukocytes and neutrophils, while others were not significant.
4. Limitations (require expansion)
The authors mention the small sample size, but it is also important to add:
• lack of data on medications and disease duration,
• age difference between groups,
• no assessment of JIA activity,
• cross-sectional design (no causal inference possible),
• lack of information on psychological support for participants.
The academic style is generally correct but requires minor linguistic corrections:
- replace “One other hand” → “On the other hand”,
- replace “RVD” → “RDW” (typographical error in table and text),
- in the abstract, “BDC” → “BDI” (typographical error),
- several sentences in the discussion repeat content from the abstract.
Author Response
Reviewer 2
- General Assessment
The article is well written, logically consistent, and aligns with the scope of the journal Children. The authors address a current and interdisciplinary topic - the relationship between depressive symptoms, sleep disturbances, and immunological markers in juvenile idiopathic arthritis (JIA). The study combines clinical, psychological, and biochemical aspects, which is undoubtedly its strength.
The research is carefully designed (power calculated in G*Power, appropriate control groups, bioethics committee approval). The conclusions are consistent with the data and literature.
However, the paper has several important limitations and areas requiring clarification, particularly in the statistical methods, interpretation of results, and linguistic accuracy.
- Strengths
- A novel and clinically important topic - few studies simultaneously assess depression, sleep, and inflammatory parameters in children with JIA.
- Appropriate choice of diagnostic tools - BDI and SDSC are valid and well-validated instruments.
- Well-developed Materials and Methods section, especially Section 2.1, which includes the power calculation.
- Results are consistent with international literature and include numerous up-to-date references (2023–2025).
- The discussion section is extensive and effectively contextualizes the findings in relation to previous studies.
Comments 1:
- Substantive Comments
Methodological Design
- There is no information about the remission or activity status of JIA at the time of the study - this is crucial, as CRP, ESR, and depression severity may depend on inflammatory activity.
Response 1:
Dear Reviewer,
First of all, thank you very much for taking the time to evaluate our manuscript and for your constructive feedback.
Taking our criticisms into account, the following sentence has been added.
“All patients included in this group were in remission.”
Comments 2: The BDI scale is not suitable for younger children (under 13 years old). Participants were aged 4–18, so the tool is inappropriate for part of the sample.
It should be indicated whether parents/guardians completed the questionnaire for younger children and how response reliability was ensured.
Response 2:
Taking your criticisms into account, the following sentences have been added.
“In this research, individuals under the age of 13 administered depression and sleep disturbance scores with parent/guardian support to increase the clarity and reliability of their responses. For children in this age group, the questionnaires were completed by their parents or guardians. During administration, the researchers guided parents to respond to each item based on their child's recent emotional state and behavior. This approach ensured the reliability and consistency of the data obtained from young children.”
Comments 3:
The control group is somewhat younger (mean age 9.18 vs. 12.36 years), which may influence sleep and mood.
This difference should be noted in the Limitations section as a potential confounding factor.
Response 3:
Taking your suggestion into consideration, the age difference was added to the limitations of the study.
Comments 4:
- The authors mention using parametric and nonparametric tests but do not specify which variables were analyzed with which test (often only “p < 0.05” is provided).
Please specify for key analyses which tests were parametric and which were nonparametric.
Response 4:
Based on your comments, tests comparing variables have been added to the appropriate places in the results section.
Comments 5:
No correction for multiple comparisons was applied, despite analyzing several SDSC subscales and biomarkers.
Please apply Bonferroni or Holm–Bonferroni correction and update the p-values accordingly.
Response 5:
To control for false positive results that may arise from multiple comparisons, Holm-Bonferroni correction was applied to the intergroup comparisons of biomarkers and SDSC subscales; however, a statement regarding this was not included in the statistical analysis section. In line with your suggestion, the following sentence regarding this issue has been added to the statistical analysis section.
“To control for false positive results that may arise from multiple comparisons, Holm-Bonferroni correction was applied to the intergroup comparisons of biomarkers and SDSC subscales.”
Comments 6:
- The correlation strength in Table 4 appears suspiciously high and likely results from a typographical error or data duplication (r = –0.998; 0.911; 0.907; 0.876, etc.).
Response 6:
Based on your criticisms, we reanalyzed the correlation analysis to correct the error in the r values. There was no change in the P values.
Comments 7:
Interpretation of Results
- The authors interpret the negative correlation between leukocytes/neutrophils and depression as “depression reduces immunological parameters.” In a cross-sectional study, the direction of the relationship cannot be established.
Please revise such formulations.
Response 7:
The relevant sentence has been corrected, taking your criticism into account.
Comments 8:
- In the discussion, it would be valuable to expand on why SHY (sleep hyperhidrosis) was lower in the JIA group, rather than merely speculating about sympathetic activity. This may result from medication use (e.g., NSAIDs or biologics).
Response 8:
In line with your suggestion, the following sentence has been added to the article.
" Additionally, children with JIA included in this study were under follow-up. The negative correlation between sleep hyperhidrosis and BDI scores may be due to medication use."
Comments 9:
The title suggests that “laboratory findings” were analyzed - it should be emphasized in the conclusion that the strongest correlations involved leukocytes and neutrophils, while others were not significant.
Response 9:
Upon your suggestion, leukocyte and neutrophil counts were added to the title instead of laboratory findings.
Comments 10:
Limitations (require expansion)
The authors mention the small sample size, but it is also important to add:
- lack of data on medications and disease duration,
- age difference between groups,
- no assessment of JIA activity,
- cross-sectional design (no causal inference possible),
- lack of information on psychological support for participants.
Response 10:
In line with your suggestion, the following sentence has been added to the article.
“Additionally, lack of data on medication use and disease duration, age differences between groups, absence of patients in the active phase, potential self-report bias, cross-sectional study design preventing causal inference, and lack of psychological support information for participants are among the important limitations of the study.”
Comments 11:
Comments on the Quality of English Language
The academic style is generally correct but requires minor linguistic corrections:
- replace “One other hand” → “On the other hand”,
- replace “RVD” → “RDW” (typographical error in table and text),
- in the abstract, “BDC” → “BDI” (typographical error),
- several sentences in the discussion repeat content from the abstract.
Response 11:
Thank you very much for helping us identify our errors. The necessary corrections have been made.
Sincerely

Reviewer 3 Report
Comments and Suggestions for Authors
This research addressed the relationship between depression and sleep disturbances, and laboratory findings in children with juvenile idiopathic arthritis (JIA)
I think the novelty of this study is quite limited. Several previous publications have already investigated the relationship between depression and disease activity, or between sleep disturbances and disease activity, in children with JIA. Although this research attempts to integrate these three aspects, I have concerns regarding the appropriateness of using the Beck Depression Inventory (BDI) for assessing depression in children. In addition, the reported laboratory findings—such as leukocyte and neutrophil levels—are not particularly convincing, as the differences appear minimal and may be influenced by various factors. Consequently, these two results have limited clinical relevance or practical significance.
It has laboratory findings in this article.
I believe the Children’s Depression Inventory would be more appropriate for this study.
The conclusions are consistent, the authors adequately addressed their main research question, although some of the information presented is already well known.
The references are appropriate.
The tables are clear, and I have no further comments regarding them.
- Is it appropriate to use the Beck Depression Inventory (BDI) to assess depression in children? Why not use the Children’s Depression Inventory (CDI) instead?
- In line 153, please note that the Kruskal–Wallis and Mann–Whitney U tests are non-parametric methods.
- Please expand the discussion to include additional limitations, such as the cross-sectional study design and potential self-report bias.
Author Response
Reviewer 3
This research addressed the relationship between depression and sleep disturbances, and laboratory findings in children with juvenile idiopathic arthritis (JIA)
Comments 1:
I think the novelty of this study is quite limited. Several previous publications have already investigated the relationship between depression and disease activity, or between sleep disturbances and disease activity, in children with JIA. Although this research attempts to integrate these three aspects, I have concerns regarding the appropriateness of using the Beck Depression Inventory (BDI) for assessing depression in children. In addition, the reported laboratory findings—such as leukocyte and neutrophil levels—are not particularly convincing, as the differences appear minimal and may be influenced by various factors. Consequently, these two results have limited clinical relevance or practical significance.
It has laboratory findings in this article.
I believe the Children’s Depression Inventory would be more appropriate for this study.
Response 1:
Dear Reviewer,
First of all, thank you very much for taking the time to evaluate our manuscript and for your constructive feedback.
Our study examining depression and sleep disturbances in children with JIA, along with laboratory findings (As you noted), is unique based on our literature review. Although the children in the case group are under follow-up and in remission, we believe that the significant laboratory findings and their correlation with depression and sleep disturbance scores provide valuable information.
Comments 2:
The conclusions are consistent, the authors adequately addressed their main research question, although some of the information presented is already well known.
The references are appropriate.
The tables are clear, and I have no further comments regarding them
Response 2:
Thank you again for your evaluation.
Comments 3:
Is it appropriate to use the Beck Depression Inventory (BDI) to assess depression in children? Why not use the Children’s Depression Inventory (CDI) instead?
Response 3:
We appreciate your comments regarding the choice of depression assessment tool. However, our study initially planned to use the Beck Depression Inventory (BDI) for all participants. While the BDI is primarily recommended for adolescents and adults, in our study, the questionnaire was administered to children under 13 years of age with the support of a parent or legal guardian to ensure clarity and reliability of responses.
Comments 4:
In line 153, please note that the Kruskal–Wallis and Mann–Whitney U tests are non-parametric methods.
Response 4:
Necessary corrections were made according to your suggestion.
Comments 5:
Please expand the discussion to include additional limitations, such as the cross-sectional study design and potential self-report bias.
Response 5:
Necessary additions have been made in accordance with your suggestion.
Sincerely

Round 2
Reviewer 2 Report
Comments and Suggestions for Authors
The authors have addressed most of the reviewers’ previous comments appropriately. The manuscript has been substantially improved in clarity, structure, and scientific precision.
However, I still have three remaining comments that should be addressed before the paper can be considered fully ready for publication:
- In the Statistical Analysis section, the authors correctly described the types of tests used and mentioned the Bonferroni correction. However, in the Results section, it is still unclear which specific tests were applied to which variables. Please explicitly state in the text which tests were used for each key comparison.
- The Limitations section mentions the age difference between the JIA and control groups, but it would be helpful to elaborate briefly on why this difference could influence sleep and mood outcomes - consistent with the reviewer’s earlier comment.
-
The addition of a sentence noting that medication use might influence the lower SHY scores in the JIA group is appreciated. However, this explanation could be expanded with concrete examples. Please provide one or two examples to strengthen this interpretation.
Overall, the manuscript has improved considerably and is close to acceptance after these minor clarifications.
Author Response
Reviewer 2
The authors have addressed most of the reviewers’ previous comments appropriately. The manuscript has been substantially improved in clarity, structure, and scientific precision.
However, I still have three remaining comments that should be addressed before the paper can be considered fully ready for publication:
Comments 1:
In the Statistical Analysis section, the authors correctly described the types of tests used and mentioned the Bonferroni correction. However, in the Results section, it is still unclear which specific tests were applied to which variables. Please explicitly state in the text which tests were used for each key comparison.
Response 1:
Dear Reviewer,
First of all, thank you very much for taking the time to evaluate our manuscript and for your constructive feedback.
In line with your suggestions, we have explicitly stated in the text which tests were used for each key comparison.
Comments 2: The Limitations section mentions the age difference between the JIA and control groups, but it would be helpful to elaborate briefly on why this difference could influence sleep and mood outcomes - consistent with the reviewer’s earlier comment.
Response 2:
Taking your suggestion into consideration, the following paragraph has been added to the article.
“A recent meta-analysis reported a consistently positive association between mental health and sleep in both adolescents and children. It also suggested that increased sleep may have a protective effect on mental health [48]. Conversely, insufficient sleep in children have been reported to lead to a variety of neurological, neurocognitive, emotional, and mood-related problems. However, sleep duration is an age-dependent variable; children of different ages require different amounts of sleep. Adequate sleep quality can be defined as an individual's ability to maintain a normal sleep pattern appropriate for their age. Therefore, the patient's age and associated etiological factors should be taken into account when planning treatment for sleep insufficiency [49]. The healthy children involved in this research were younger than the children with JIA.”
Comments 3:
The addition of a sentence noting that medication use might influence the lower SHY scores in the JIA group is appreciated. However, this explanation could be expanded with concrete examples. Please provide one or two examples to strengthen this interpretation.
Response 3:
Taking your suggestion into consideration, the following paragraph has been added to the article.
“Additionally, children with JIA included in this study were under follow-up and in remission. The negative correlation between sleep hyperhidrosis and BDI scores is thought to be due to the anti-inflammatory impacts of the medications used. The treatment's ability to control inflammation provides a plausible basis for the indirect reduction in symptoms such as sleepiness and sweating [38]. Indeed, a study by Khubchandani and colleagues reported improvement in sleep hyperhidrosis after treatment in children with active disease [39].”
Comments 4:
Overall, the manuscript has improved considerably and is close to acceptance after these minor clarifications.
Response 4:
Dear Reviewer,
Thank you very much for your positive comments and constructive feedback. We have carefully considered all three suggested corrections and revised the article accordingly.
Sincerely

Reviewer 3 Report
Comments and Suggestions for Authors
I am satisfied with the author's reply, I have no further comments.
Author Response
Reviewer 3
I am satisfied with the author's reply, I have no further comments.
Response :
Dear Reviewer,
Thank you very much for taking the time to evaluate our manuscript and for your constructive feedback.
Sincerely
